# Systemic Oxidative Stress Markers in Cirrhotic Patients with Hepatic Encephalopathy: Possible Connections with Systemic Ammoniemia

**DOI:** 10.3390/medicina56040196

**Published:** 2020-04-23

**Authors:** Cătălin Sfarti, Alin Ciobica, Ioana-Miruna Balmus, Ovidiu-Dumitru Ilie, Anca Trifan, Oana Petrea, Camelia Cojocariu, Irina Gîrleanu, Ana Maria Sîngeap, Carol Stanciu

**Affiliations:** 1Department of Gastroenterology, “Grigore T. Popa” University of Medicine and Pharmacy, “St. Spiridon” University Hospital, Institute of Gastroenterology and Hepatology, Independence Avenue, no 1, 700111 Iași, Romania; cvsfarti@gmail.com (C.S.); ancatrifan@yahoo.com (A.T.); stoica_oanacristina@yahoo.com (O.P.); gilda_iri25@yahoo.com (I.G.); anamaria.singeap@yahoo.com (A.M.S.); 2Department of Research, Faculty of Biology, “Alexandru Ioan Cuza” University, Carol I Avenue, no 11, 700505 Iasi, Romania; balmus.ioanamiruna@yahoo.com (I.-M.B.); ovidiuilie90@yahoo.com (O.-D.I.); 3Department of Interdisciplinary Research in Science, Alexandru Ioan Cuza University of Iasi, Carol I Avenue, no. 11, 700506 Iasi, Romania; 4Romanian Academy, Iasi Branch, Nr. 8, Carol I Avenue, no. 8, 700505 Iasi, Romania; stanciucarol@yahoo.com

**Keywords:** liver cirrhosis, glutathione peroxidase, malondialdehyde, superoxide dismutase, ammoniemia

## Abstract

*Background and objectives:* Oxidative stress shows evidence of dysregulation in cirrhotic patients with hepatic encephalopathy (HE), although there are still controversies regarding the connections between oxidative stress and ammonia in these patients. The aim of this study was to evaluate the oxidative stress implication in overt HE pathogenesis of cirrhotic patients. *Materials and Methods:* We performed a prospective case-control study, which included 40 patients divided into two groups: group A consisted of 20 cirrhotic patients with HE and increased systemic ammoniemia, and group B consisted of 20 cirrhotic patients with HE and normal systemic ammoniemia. The control group consisted of 21 healthy subjects matched by age and sex. The activity of superoxide dismutase (SOD), glutathione peroxidase (GPx), malondialdehyde (MDA) levels (lipid peroxidation marker), and ammoniemia were evaluated. *Results*: We found a significant decrease in SOD and GPx activity and also a significant increase of MDA levels in cirrhotic patients with HE as compared to the healthy age-matched control group (1.35 ± 0.08 vs. 0.90 ± 0.08 U/mL, *p* = 0.002; 0.093 ± 0.06 vs. 0.006 ± 0.008 U/mL, *p* = 0.001; and 35.94 ± 1.37 vs. 68.90 ± 5.68 nmols/mL, *p* = 0.0001, respectively). Additionally, we found significant correlations between the main oxidative stress markers and the levels of systemic ammonia (r = 0.452, *p* = 0.005). Patients from group A had a significant increase of MDA as compared with those from group B (76.93 ± 5.48 vs. 50.06 ± 5.60 nmols/mL, *p* = 0.019). Also, there was a compensatory increase in the activity of both antioxidant enzymes (SOD and GPx) in patients with increased systemic ammoniemia (group A), as compared to HE patients from group B. *Conclusions:* Our results demonstrated a significant decrease in antioxidants enzymes activities (SOD and GPx), as well as a significant increase in MDA concentrations, adding new data regarding the influence of oxidative stress in HE pathogenesis in cirrhotic patients.

## 1. Introduction

Hepatic encephalopathy (HE) is currently one of the main complications of liver cirrhosis and a characteristic of acute liver failure. Once developed, this neuropsychiatric manifestation of chronic or acute liver disease is associated with high mortality and healthcare costs [1].

HE is commonly defined as a brain dysfunction secondary to liver insufficiency and/or portosystemic shunting that manifests as a wide spectrum of neurological or psychiatric abnormalities ranging from subclinical disturbances to coma [2].

Due to its multifactorial, complex and yet unknown pathophysiology, it is thought that HE is generally caused by cerebral edema as a result of the combined action of several factors such as neurotoxins (e.g., ammonia—NH_3_), impaired neurotransmission following metabolic changes in liver failure, alteration of blood-brain barrier permeability, systemic inflammation, and also oxidative and nitrosative stress [2,3].

Although ammonia is widely recognized as the main factor involved in the pathogenesis of HE, the correlation between ammoniemia and HE severity was shown to be significant in acute liver failure by contrast to chronic liver disease. This could suggest that the latter could be influenced also by other pathological factors [4]. 

The systemic inflammation associated with liver cirrhosis that can trigger the production of reactive oxygen species (ROS) is also able to generate astrocyte swelling and rapid deterioration of neuropsychological function [5].

There is an increasing number of studies in various animal models that described a possible relevance of oxidative stress as a pathogenic mechanism involved in HE [6,7,8,9,10]. Although, the implications of oxidative stress in HE pathophysiology are not completely demonstrated, it is clear that there has been an increasing trend in the last 20 years to consider oxidative stress as an important trigger of HE [4,5].

The aim of our study was to evaluate the role of oxidative stress in the pathogenesis of HE in cirrhotic patients and to establish possible correlations between the systemic oxidative stress markers and the serum ammonia levels. 

## 2. Material and Methods

### 2.1. Patients and Healthy Subjects

We conducted a prospective case-control study on cirrhotic patients admitted in a tertiary hospital between April 2018 and June 2018. The study included Child–Pugh class B (15 patients) or C (25 patients) cirrhotic patients, with newly diagnosed overt HE (n = 40, 19 males and 21 females, mean age 56.0 ± 10.4 years). 

Exclusion criteria were diagnosis of a neurological or psychiatric disease, anterior diagnosis and/or treatment of HE, current use of psychotropic medications, as well as the absence of signed informed consent. Subjects taking antioxidant supplements were also excluded and also the patients with clinical or biological signs of infection were not included in the study.

The cirrhotic patients were classified according to ammoniemia in two groups. Group A—20 cirrhotic patients with HE and increased systemic ammoniemia and group B—20 cirrhotic patients with HE and normal systemic ammoniemia. The control group consisted of 21 healthy subjects (10 males and 11 females, aged 49.8 ± 9.76 years) matched by age and sex.

The following data have been collected for all patients: demographics, cirrhosis etiology, Child–Pugh class and Model for End-Stage Liver Disease (MELD) score, medication (used for prevention/treatment of liver cirrhosis and its complications), ammoniemia, oxidative stress markers, HE evaluation (assessed using the West Haven mental status scale). The baseline characteristics of the study groups are presented in Table 1.

All subjects or their caregivers gave their informed consent for inclusion before they participated in the study. The study was conducted in accordance with the Declaration of Helsinki, and the protocol was approved by the Ethics Committee of “Grigore T. Popa” University of Medicine and Pharmacy of Iasi (no. 19856/03.02.2018).

### 2.2. Measurement of Oxidative Stress Markers

Blood samples were collected following an overnight fasting period during the morning hours at the time of admission and allowed to clot. After centrifugation (3000 rpm, 15 min, 4 °C), blood sera were separated, aliquoted and stored at −80 °C until subsequent analysis. Serum samples were used to determine serum levels of the oxidative stress markers.

SOD activity was measured in a direct dependency manner with reaction inhibition rate (%) of WST-1 substrate (a water-soluble tetrazolium dye) with xanthine oxidase using a SOD Assay Kit 19160 (Merck, Darmstadt, Germany) from a 40 µl serum sample, according to the manufacturer’s instructions. Each endpoint assay was monitored by absorbance at 450 nm (the absorbance wavelength for the colored product of WST-1 reaction with superoxide) (PharmaSpec UV-1700, Shimadzu, Kyoto, Japan) after 20 min of reaction time at 37 °C.

Cellular GPx activity was measured from a 50 µl serum sample using the GPx cellular activity assay kit CGP-1 (Merck, Darmstadt, Germany). The Glutathione Peroxidase Assay Kit method used an indirect quantification based on coupled enzymatic reactions (GPx-oxidized glutathione-reduced glutathione—glutathione reductase enzymatic chain). During the in vitro assay reactions, the GPx-catalyzed glutathione oxidation is coupled with the glutathione-reductase-catalyzed reverse reaction. Thus, the NADPH consumption can be quantified and due to its direct proportionality, GPx activity can be calculated (PharmaSpec UV-1700 spectrophotometer, Shimadzu, Japan).

Antioxidant enzymes activities were normalized against serum total soluble protein concentration and expressed as enzymatic specific activity units. Therefore, total soluble protein content was measured from 0.2 mL serum sample using Bradford spectrophotometric assay method (PharmaSpec UV-1700, Shimadzu, Japan), in which absorbance at 595 nm was calculated against bovine serum albumin curve, and the results were expressed as mg total soluble proteins/mL serum. 

MDA levels were determined by thiobarbituric acid-reactive substances assay. Serum at a quantity 0.2 mL was added and briefly mixed with 1 mL of trichloroacetic acid at 50%, 0.9 mL of TRIS–HCl (pH 7.4) and 1 mL of thiobarbituric acid 0.73%. Following vortex mixing, samples were maintained at 100 °C for 20 min. Afterward, the samples were centrifuged at 3000 rpm for 10 min and supernatant read at 532 nm (PharmaSpec UV-1700, Shimadzu, Japan). The signal was calculated against an MDA standard curve and the results were expressed as nmols/mL.

Measurement of systemic ammoniemia.

Ammoniemia was measured from fresh EDTA-enriched blood samples following centrifugation at 4 °C in an integrated automated chemistry system. The ammonia assay reagent is based on glutathione dehydrogenase reaction using a stable analog of NADPH. The analysis was performed according to the manufacturer’s instructions and the results were automatically calculated and expressed as µmols/mL. The threshold of high ammonemia was any value above the normal threshold of the hospital laboratory 30–120 mcg/dL. In this regard, the normal ammonemia was considered any value below 120 mcg/dL and high ammonemia all values above 120 mcg/dL.

### 2.3. Statistical Analysis

All statistical tests were performed using the Statistical Package for the Social Sciences software (SPSS version 20; SPSS, Chicago, IL, USA). The Kolmogorov test was used to test the pattern for normal distribution. Continuous variables were expressed as median (interquartile range) and the Mann–Whitney U test or Chi-square test was used to compare parameters. To compare the groups, Student’s *t*-test was used for the variables with normal distribution, and the Mann–Whitney U test was used for the variables with asymmetric distribution. Comparisons were carried out by analysis of variance (ANOVA). Correlations between oxidative stress-related markers and serum ammonia levels were analyzed using the Spearman’s rank correlation method. Differences were considered statistically significant when the *p*-value was less than 0.05.

## 3. Results

### 3.1. Patient Characteristics

This study included 40 patients with liver cirrhosis and HE that were classified into subgroups, according to the serum ammonia level (group A—increased systemic ammoniemia and group B—normal systemic ammoniemia) and 21 participants in the control group.

Demographic, clinical and laboratory characteristics of all patients, as well as of those with increased or normal systemic ammoniemia are shown in Table 1. Baseline characteristics were generally similar in the two groups. From the liver diseases patients, 24 (60.0%) patients had alcoholic ethology, and 16 (40.0%) had viral etiology (10 with hepatitis C virus-related cirrhosis, and 6 with hepatitis B virus-related cirrhosis). Overall, the median total bilirubin level was 4.03 mg/dL and the median MELD score was 15.0. There were no significant differences between cirrhosis patients with HE and high ammoniemia (group A) and those with normal ammoniemia (group B) regarding gender distribution, cirrhosis etiology, liver disease severity (MELD score, Child–Pugh score), and monitored cirrhosis complications (ascites, spontaneous bacterial peritonitis, upper gastrointestinal bleeding and hepato-renal syndrome), as well as most of the laboratory parameters (Table 1). However, group A was characterized by significantly more frequent grade IV hepatic encephalopathy (35.0% vs. 10.0%, *p* = 0.043). 

### 3.2. Oxidative Stress-Related Markers 

Initial analysis of our results included all liver cirrhosis and newly diagnosed overt HE patients, regardless of their systemic ammoniemia. Biochemical data showed a significant decrease of SOD activity in all subjects with HE, as compared to control group (Table 2) (0.90 ± 0.08 vs 1.35 ± 0.08 U/mL, *p* = 0.002).

The activity of GPx, was significantly decreased in the HE group compared to healthy age and sex-matched subjects (Table 2) (0.061 ± 0.008 vs. 0.09 ± 0.006 U/mL, *p* = 0.01). In addition, we found that lipid peroxidation marker (MDA) levels were significantly increased (*p* = 0.0001) in the serum of the HE patients, as compared to control group (Table 2) (76.93 ± 5.48 vs. 35.94 ± 1.37 nmols/mL, *p* = 0.0001).

While analyzing the groups separately, by considering systemic ammoniemia (group A and group B), our results showed significant differences in terms of systemic SOD activity (1.06 ± 0.07 in group A vs. 0.68 ± 0.08 U/mL in group B, *p* = 0.0002), suggesting that ammonia could modulate enzymatic activity of SOD (Table 2).

In this way, statistical analysis revealed significant SOD activity increase in HE patients with normal ammoniemia, as compared to controls (0.90 ± 0.08 vs. 1.35 ± 0.08, *p* < 0.0001), while in HE patients with high ammoniemia SOD activity was decreased, as compared to the control group (1.06 ± 0.07 vs. 1.35 ± 0.08 U/mL, *p* = 0.033) (Table 2). Moreover, our results suggested a significant increase of SOD activity in HE patients and high ammoniemia, as compared to HE patients with normal ammoniemia (Table 2). 

Serum GPx activity was significantly different in all three groups (F(2.58) = 14, *p* < 0.0001)) (Table 2). Our results analysis showed a significant GPx activity decrease in HE patients with normal ammoniemia, as compared to controls (0.024 ± 0.004 vs. 0.09 ± 0.006 U/mL, *p* < 0.0001). No significant differences regarding GPx activity were obtained while comparing HE patients with high ammoniemia to healthy age and sex-matched controls (0.08 ± 0.005 vs. 0.09 ± 0.006 U/mL, *p* = 0.41) (Table 2). However, significant increases of GPx activity were noted in HE patients with increased ammoniemia, as compared to HE patients with normal ammoniemia (0.08 ± 0.005 vs. 0.024 ± 0.004 U/mL, *p* = 0.0008) (Table 2). 

Regarding the serum MDA levels, the main lipid peroxidation biochemical marker, the initial analysis showed significant group differences (F(2.58) = 14, *p* < 0.0001)). Thus, statistical analysis revealed a significant increase of MDA levels in HE patients with normal ammoniemia, as compared to controls (50.06 ± 5.6 vs 35.94 ± 1.37 nmols/mL, *p* = 0.007), as well as a significant increase of MDA concentration in HE patients with high ammoniemia, as compared to control group (76.93 ± 5.48 vs. 35.94 ± 1.37 nmols/mL, *p* < 0.0001) (Table 2). 

In addition, a significant increase of serum MDA concentrations were obtained in HE patients with high ammoniemia, as compared with HE patients with normal ammoniemia (76.93 ± 5.48 vs. 50.06 ± 5.60 nmols/mL, *p* = 0.017) (Table 2).

### 3.3. Correlations between Oxidative Stress-Related Markers and Serum Ammonia Level

Pearson correlation coefficient analysis for all HE patients showed significant direct correlations between the two antioxidant enzymes as follows: SOD vs. ammonia levels (n = 40, r = 0.452, *p* = 0.005) (Figure 1A), and GPx vs ammonia levels (n = 40, r = 0.460, *p* = 0.007) (Figure 1B). In addition, we obtained a strong positive correlation between MDA levels and ammoniemia (n = 40, r = 0.855, *p* = 0.0001) (Figure 1C). However, no significant correlations were found while comparing inflammation marker C reactive protein with ammoniemia (n = 40, r = –0.078, *p* = 0.733).

## 4. Discussion

In the present study, we found an increased oxidative stress in HE engaged by the significant decrease of serum antioxidant enzymatic activity (SOD and GPx), as well as by the significant increase of MDA concentration, as a lipid peroxidation marker. Moreover, our results are indicating important correlations between systemic ammoniemia and all of the evaluated oxidative stress markers (SOD, GPx and MDA). Also, while evaluating the patients based on their ammoniemia, significant increases of MDA levels were obtained for HE patients with increased ammoniemia, as compared with those with normal ammoniemia. In addition, the increased systemic ammoniemia in HE associated with the present antioxidant enzymes activity trending could suggest the compensatory effect in which, as a result of high oxidative stress (suggested by MDA levels), the antioxidant enzymes try to overcome free radical accumulation and subsequent damage (Figure 1).

However, in normal ammonia levels cirrhotic patients, a high frequency of severe forms (III—55% and IV—10%) was recorded. In this case, not the hyperammonemia theory but the oxidative stress theory could explain the increased occurrence of neurological symptoms. Under normal circumstances as has been portrayed in Figure 2, approximatively 90% of the total oxygen is taken up by the mitochondria in order to produce ATP, ROS resulting through the partial four-electron reduction of O_2_ to H_2_O [11]. When the metabolic rate is increased, this is defined by a high production of ROS and is reflected by a depletion of the main antioxidant enzymes, in the present case, SOD and GPx [12]. However, the only reliable possibility to decrease ROS generation is to uncouple the metabolic rate [13].

Regardless of the status of the organism, ROS gradually promotes a dysbiosis [14], apart from the imperative mitochondrial dysfunction that exacerbates ROS production [15]. Once the eubiosis is lost, a hazardous environment within the gut microflora emerges and disturbs its entire metabolism [16,17].

Subsequently, a series of pro-inflammatory cytokines are activated that further alter protein metabolism [18]. Accordingly, the levels of MDA are increased, NH_3_ is released and through the portal vein circulates to the liver [2].

NH_3_ enters the systemic circulation and when the levels are higher than 500 µM, systemic inflammation of the glial cells occurs, the blood–brain barrier being unable to act efficiently, which ultimately leads to hepatic encephalopathy [19]. It must be taken into considerations that this mechanism applies by excluding all harmful exogenous factors.

In our study, we obtained similar results regarding the increased serum MDA concentration, used to assess lipid peroxidation intensity.

An important mechanism that could be described in this context was correlated with hyperamonemia-induced oxidative stress, as some previous studies suggested [20,21]. Thus, it was shown that the stimulation of N-methyl-D-aspartate receptors by ammonia could result mainly in antioxidant enzymes activity decreasing and superoxide or hydroxyl radicals levels increasing [5]. Also, acute astrocytes edema can be implicated in this context, while they could be modulated by the reactive oxygen species and reactive nitrogen species [22].

Moreover, regarding the relation between ammonia and oxidative stress, the well-known “two-hit hypothesis” of HE could be mentioned. Thus, it states that increased ammonia, followed by astrocytes dysfunction and oxidative stress is the initial hit. Afterward, the second hit is represented by gastrointestinal bleeding, infection or dehydration-induced ammonia load [7].

In the present report, we showed that together with the increased oxidative stress status in HE patients’ blood serum, strong correlations between systemic oxidative stress markers (two antioxidant enzymes-SOD and GPx and one lipid peroxidation marker-MDA) and ammoniemia were obtained. Moreover, we hereby managed to separately describe the HE patients based on ammoniemia and to report a significant increase of MDA levels in HE patients with increased ammoniemia, as compared to HE patients bearing normal ammoniemia. 

Similar studies demonstrated that the increased oxidative stress status is not characteristic to systemic level in HE, as we hereby discuss based on our results, but rather to brain tissues, as in the case of the Gorg et al. study [21] that showed that some post-mortem brain cortical tissue oxidative stress markers are increased in HE patients [23]. While these oxidative modifications were mainly found in cerebral proteins and cortical RNA, no significant changes of both manganese and copper-dependent SOD activity were recorded.

In our study, we also showed some variations in SOD activity. Whereas it was significantly decreased in HE patients, as compared to controls, we demonstrated that together with GPx it exhibits a compensatory increase by direct correlation to ammoniemia in HE patients with increased ammoniemia, as compared to HE patients with normal ammoniemia. This was also previously reported by the Singh group [24] who described the increased compensatory SOD activity after chronic ammonia-exposure in animal models of HE. Due to the fact that SOD and GPx are critical first-line antioxidant enzymes, it is currently known that they act cooperatively at different sites in the metabolic pathway of free radicals, and could perhaps act as compensatory regulation in response to increased oxidative stress [25,26].

While our study strengths were already described and discussed, we should also mention some limitations. Thus, in our best of knowledge, it is only the second prospective study evaluating the influence of serum oxidative stress markers on hospitalized cirrhotic patients with HE in a Romanian tertiary referral center. However, the fact that our study was conducted in single center range by evaluating a small number of patients could be a bias source. Notwithstanding these limitations, we believe our study brings relevant evidence for investigators evaluating the role of oxidative stress in the pathogenesis of hepatic encephalopathy. In addition, we can mention here the lack of a description of spontaneous portosystemic shunts, which could have helped to explain non hyperammonemic encephalopathy.

## 5. Conclusions

Our results demonstrated the increased systemic oxidative stress in HE cirrhotics patients, as shown by the significant decrease of both antioxidants enzymes activity, SOD and GPx, as well as by the significant increase of MDA serum levels. Significant correlations between the ammoniemia and all three markers of oxidative stress (SOD, GPx, MDA) were obtained. Also, based on the systemic ammonia levels of the HE patients group division, we observed a significant MDA increase in HE patients with increased ammoniemia, as well as a compensatory increase in the activity of both antioxidant enzymes (SOD and GPx), as compared with HE patients bearing normal ammoniemia. Further research is necessary in order to elucidate the role of oxidative stress in the pathogenesis of HE in cirrhotic patients. 

## Figures and Tables

**Figure 1 medicina-56-00196-f001:**
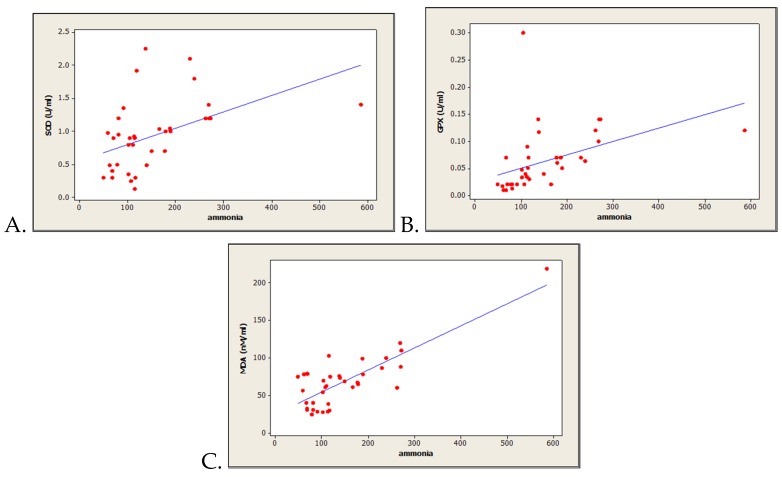
Correlations between oxidative stress markers versus systemic ammonia levels in controls and HE patients (n = 61). (**A**) Superoxide dismutase vs. systemic ammonia levels; (**B**) glutathione peroxidase vs. systemic ammonia levels; (**C**) malondialdehyde vs. systemic ammonia levels.

**Figure 2 medicina-56-00196-f002:**
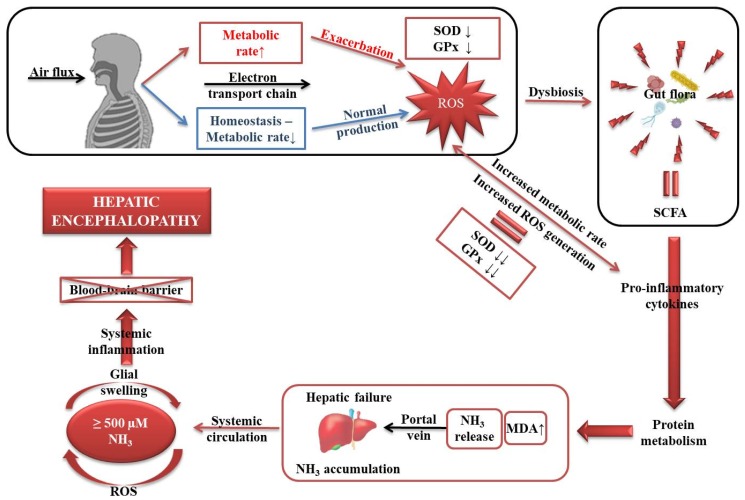
Schematic representation of how oxidative stress gradually disrupts the homeostasis and promotes hepatic encephalopathy.

**Table 1 medicina-56-00196-t001:** Patient demographics, clinical and laboratory parameters of all patients, and according to study groups.

Parameter	All Patients n = 40	Study Group A (High Ammoniemia) n = 20	Study Group B (Normal Ammoniemia) n = 20	*P*-Value
Gender, Male/Female (%)	19/21 (47.5/52.5)	11/9 (55/45)	8/12 (40/60)	0.342 ^C^
Age, Years, Mean ± SD	56.0 ± 10.4	54.3 ± 9.6	57.6 ± 10.4	0.305 ^T^
Etiology of Cirrhosis, n (%)				0.322 ^F^
HCV	10 (25.0)	3 (15.0)	7 (35)	
HBV	6 (15.0)	3 (15.0)	3 (15.0)	
Alcohol	24 (60.0)	14 (70.0)	10 (50.0)	
Child–Pugh Class B/C, n (%)	15/25 (37.5/62.5)	4/16 (20/80)	11/9 (55/45)	0.022 ^F^
Child–Pugh Ccore, Median (Q1/Q3)	10 (8/15)	10.5 (9/15)	9(8/13)	0.077 ^M^
MELD Score, Median (Q1/Q3)	15 (11/20)	21 (11/35)	17 (8/32)	0.356 ^M^
Creatinine (mg/dL),Median (Q1/Q3)	0.78 (0.11/3.5)	0.85 (0.12/2.77)	0.73 (0.11/3.5)	0.810 ^M^
Albumin (g/L), Median (Q1/Q3)	2.8 (1.59/4.5)	2.9 (1.5/3.5)	2.6 (1.96/4.5)	0.394 ^M^
Bilirubin (mg/dL), Median (Q1/Q3)	4.03 (1.74/24.6)	2.15 (1.73/24.1)	4.62 (1.8/16.0)	0.121 ^M^
INR, Median (Q1/Q3)	1.5 (1.28/1.74)	1.42 (1.22/1.63)	1.5 (1.3/1.78)	0.461 ^M^
Ammonia (µmols/L), Median (Q1/Q3)	117 (91.5/188)	158 (113.5/234.5)	69 (60.5/79.5)	0.001 ^M^
Ascites, Mild/Medium/ Large, n (%)	10/14/16 (25/35/40)	5/8/7 (25/40/35)	5/6/9 (25/30/45)	0.536 ^F^
SBP, n (%)	6 (15.0)	2 (10.0)	4 (20.0)	0.096 ^F^
HRS, n (%)	5 (12.5)	3 (15.0)	2 (10.0)	0.633 ^F^
UGIB, n (%)	13 (32.5)	6 (30.0)	7 (35.0)	0.736 ^F^
Encephalopathy, n (%)				0.043 ^F^
Stage II	16 (40.0)	9 (45.0)	7 (35.0)	
Stage III	15 (37.5)	4 (20.0)	11 (55.0)	
Stage IV	9 (22.5)	7 (35.0)	2 (10.0)	

Abbreviations: HBV, chronic hepatitis B virus; HCV, chronic hepatitis C virus; HRS, hepatorenal syndrome; INR, International Normalized Ratio; MELD, Model for End-Stage Liver Disease; SBP, spontaneous bacterial peritonitis; SD, standard deviation; UGIB, upper gastrointestinal bleeding. ^C^ Chi-square test; ^F^ Fisher exact test; ^T^ The t-test (student test); ^M^ Mann–Whitney U Test (nonparametric test).

**Table 2 medicina-56-00196-t002:** Oxidative stress markers according to study groups.

Parameter	Controls n = 21	All Cirrhotic Patients n = 40	Study Group A (High Ammoniemia) n = 20	Study Group B (Normal Ammoniemia) n = 20
SOD (U/mL)	1.35 ± 0.08	0.90 ± 0.08	1.06 ± 0.07 ª	0.68 ± 0.08
GPx (U/mL)	0.09 ± 0.006	0.061 ± 0.008	0.08 ± 0.005 ª	0.024 ± 0.004
MDA (nmols/mL)	35.94 ± 1.37	68.90 ± 5.68	76.93 ± 5.48	50.06 ± 5.60

Abbreviations: MDA, malondialdehyde; SOD, superoxide dismutase; GPx, glutathione peroxidase; ª —compensatory increase, as compared to group B.

## Data Availability

The datasets used to support the findings of this study are available from the corresponding author upon request.

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
