# Peer review of "Systemic Oxidative Stress Markers in Cirrhotic Patients with Hepatic Encephalopathy: Possible Connections with Systemic Ammoniemia"

_medicina, 2020, doi:10.3390/medicina56040196_

Round 1

Reviewer 1 Report

Medicina

Systemic oxidative stress markers in cirrhotic patients with hepatic encephalopathy: possible connections with systemic ammoniemia

In this manuscript, Sfarti et al found a significant decrease in antioxidants enzymes activities (SOD and GPx) as well as a significant increase in MDA concentrations in patients with liver cirrhosis and hepatic encephalopathy (HE), suggesting the influence of oxidative stress in HE pathogenesis in cirrhotic patients. There are, however, some issues to be clarified:

Major comments

  1. The major concern is the lack of novelty: It has been well known that mediate the synergistic effects of hyperammonemia and inflammation on cognitive and motor impairment in HE (Gimenez-Garzó C et al., Antioxid Redox Signal 2015;22:871-7.). Furthermore, the systemic inflammation associated with liver cirrhosis can trigger the production of reactive oxygen species (ROS), leading to astrocyte swelling and rapid deterioration of neuropsychological function (Cichoż-Lach H et al., World J Gastroenterol 2013;19:26-34.). In addition, hyperamonemia–induced oxidative stress has also been noticed (Haussinger D et al., Curr Opin Clin Nutr Metab Care 2010;13:87–92.; Gorg B et al., Hepatology 2008;48:567-9.). Therefore, it is not new to show blood oxidative stress markers and ammonia in cirrhotic patients with HE.
  2. 3.3. Correlations between oxidative stress-related markers and serum ammonia level: although some oxidative stress markers levels correlate with ammonia levels, is it possible that both oxidative stress and ammonia are linked to HE, rather than a causal relationship between oxidative stress markers and ammonia levels?
  3. Discussion, the 1st paragraph: the interpretation of data is not quite convincing. Furthermore, the study led by Singh et al. [11] evaluated brain but not blood levels of antioxidant enzymes activity.

Minor comments

  1. 2.1. Patients and healthy subjects: Were patients with recent or ongoing infection should excluded? Infection will influence the oxidative stress marker levels.
  2. Table 1: It is not clear why Study group B (normal ammoniemia) patients, as compared with Study group A (high ammoniemia), had a higher percentage of patients with stage III HE.

Author Response

Reviewer  1

Major comments

  1. The major concern is the lack of novelty: It has been well known that mediate the synergistic effects of hyperammonemia and inflammation on cognitive and motor impairment in HE (Gimenez-Garzó C et al., Antioxid Redox Signal 2015;22:871-7.). Furthermore, the systemic inflammation associated with liver cirrhosis can trigger the production of reactive oxygen species (ROS), leading to astrocyte swelling and rapid deterioration of neuropsychological function (Cichoż-Lach H et al., World J Gastroenterol 2013;19:26-34.). In addition, hyperamonemia–induced oxidative stress has also been noticed (Haussinger D et al., Curr Opin Clin Nutr Metab Care 2010;13:87–92.; Gorg B et al., Hepatology 2008;48:567-9.). Therefore, it is not new to show blood oxidative stress markers and ammonia in cirrhotic patients with HE.

Response: Thank you for your kind and relevant concern. There was no statement that this paper is the first research in the field, our purpose being to “add new data regarding the influence of oxidative stress in HE pathogenesis in cirrhotic patients”. On the other hand, the published studies in this field are rare and the pathogenesis of hepatic encephalopathy is based on theories which are not all completely proven as Cichoż-Lach H et al states in his paper “Although the etiology of HE has not been conclusively established, it is widely agreed to be the result of numerous pathological processes. Research on the pathogenesis of HE is still being conducted, and several theories have been reported”. With that being said, we have decided to start and conduct the study in the direction of the oxidative stress-related processes. An additional figure (required by the other reviewer of the manuscript), presenting some of these possible correlations was also added in the discussion section. Some additional text was also added to cover the aforementioned aspects and the description of the Figure 2 in the Discussion section:

“Under normal circumstances as has been portrayed in Figure 2, aproximatively 90% of the total oxygen is taken up by the mitochondria in order to produce ATP, ROS resulting through the partial four-electron reduction of O2 to H2O [11].

When the metabolic rate is increased, this is defined by a high production of ROS and is reflected by a depletion of the main antioxidant enzymes, in the present case, SOD and GPx [12]. However, the only reliable possibility to decrease ROS generation is to uncouple the metabolic rate [13].

Regardless the status of the organism, ROS gradually promotes a dysbiosis [14], apart from the imparative mitochondrial dysfunction which exacerbate ROS production [15]. Once the eubiosis is lost, a hazardous environment within the gut microflora emerges and disturb its entire metabolism [16,17].

Subsequently are activated a series of pro-inflammatory cytokines which further alter the protein metabolism [18]. Accordingly, the levels of MDA are increased, NH3 is released and through the portal vein circulate to the liver [2].

NH3 enters the systemic circulation and when the levels are higher than 500 µM, it occurs a systemic inflammation of the glial cells, the blood-brain barrier being unable to act efficiently which ultimately leads to hepatic encephalopathy [19].

It must be taken into considerations that this mechanism applies by excluding all harmful exogenous factors.”

Thank you!

Figure 2. Schematic representation on how oxidative stress gradually disrupts the homeostasis and promote hepatic encephalopathy. Given the fact that humans are strictly aerobic organisms, ROS results through the partial four-electron reduction of O2 to H2O. When the metabolic rate is increased, this is defined by a high production of ROS and vice versa and is reflected by a decreased of SOD and GPx. ROS promotes a dysbiosis, creating a hazardous environment in the gut microflora which disturb its entire metabolism. Subsequently are activated a series of pro-inflammatory cytokines which further alter the protein metabolism. Accordingly, it occurs the accumulation of MDA, NH3 is released and through the portal vein circulate to the liver. NH3 enters the systemic circulation and when the levels are higher than 500 µM, a system inflammation of the glial cells is installed. The blood-brain barrier no longer is able to act efficiently and ultimately leads to hepatic encephalopathy. It must be taken into considerations that this mechanism applies by excluding all harmful exogenous factors.

  1. 3.3. Correlations between oxidative stress-related markers and serum ammonia level: although some oxidative stress markers levels correlate with ammonia levels, is it possible that both oxidative stress and ammonia are linked to HE, rather than a causal relationship between oxidative stress markers and ammonia levels?

Response: Thank you again for your kind and relevant concern. We feel that the significant differences of stress-related markers between group A and group B (high or normal ammonia levels) suggest most likely a causal effect rather than just a random link of both parameters to HE. Again, an additional figure, presenting some of these possible correlations was added in the discussion section and the aforementioned new text in the Discussion section. Thank you for your suggestion.

  1. Discussion, the 1st paragraph: the interpretation of data is not quite convincing. Furthermore, the study led by Singh et al. [11] evaluated brain but not blood levels of antioxidant enzymes activity.

Thank you for drawing our attention into that matter. The paragraph was eliminated due to the fact that it does not present a reasonable argument for the context of the discussion.

 Minor comments

  1. 2.1. Patients and healthy subjects: Were patients with recent or ongoing infection should excluded? Infection will influence the oxidative stress marker levels.

Response: This is a very good observation and during the patient selection, we took into account the presence of ongoing infection. The patients with clinical or biological signs of infection were not included in the study, but it was not stated as an exclusion criterion, so we did add it now.

  1. Table 1: It is not clear why Study group B (normal ammoniemia) patients, as compared with Study group A (high ammoniemia), had a higher percentage of patients with stage III HE.

Response: Thank you for suggestion this unclear aspect of our manuscript. As we stated in the introduction of this paper, in cirrhosis, there is not always a clear correlation between ammoniemia and HE severity and this was one of the main reasons of starting this study: to find out if oxidative stress might be a supplementary etiological factor of HE. “Although ammonia is widely recognized as the main factor involved in the pathogenesis of HE, the correlation between ammoniemia and HE severity was shown to be significant in acute liver failure by contrast to chronic liver disease. This could suggest that the latter could be influenced also by other pathological factors”. Thank you!

Reviewer 2 Report

Sfarti et al. provide a prospective study evaluation the oxidative stress status in regard to ammonemia level in cirrhotic patients.

Although the link between inflammation/oxidative stress and encephalopathy, the authors nicely analyze the correlation between ammonemia level and SOD, GPx and MDA. Interestingly, they show that oxidative stress is associated with encephalopathy without hyperammonemia but more severe encephalopathy appears when hyperammonemia occurs.

It is not clear in the manuscript how the authors interpret this result. Is it oxidative stresss that leads to hyperammonemia or hyperammonemia that leads to further oxidative stress although it exists "a compensatory increase" in GPx ? This should be clarified ; a figure could be helpful.

Additionnal comments

- there is no threshold for "high ammonemia" (this can vary according to the lab). Authors should precise this parameter that define their groups

- a description of spontaneous portosystemic shunts may help to explain non hyperammonemic encephalopathy

- furthermore, the authors should discuss more thoroughly this group of non-hyperammonemic encephalopathy in the light of their findings

- in this view, the discussion is too long and repeats some of the data detailed in the introduction and concentrate on the previous comments.

I thank the authors and the editor for the opportunity to review this manuscript. 

Author Response

Reviewer  2

Although the link between inflammation/oxidative stress and encephalopathy, the authors nicely analyze the correlation between ammonemia level and SOD, GPx and MDA. Interestingly, they show that oxidative stress is associated with encephalopathy without hyperammonemia but more severe encephalopathy appears when hyperammonemia occurs.

It is not clear in the manuscript how the authors interpret this result. Is it oxidative stresss that leads to hyperammonemia or hyperammonemia that leads to further oxidative stress although it exists "a compensatory increase" in GPx ? This should be clarified; a figure could be helpful.

Thank you for your kind suggestion. An additional figure, presenting some of these possible correlations was added in the Discussion section. Some additional text was also added to cover the aforementioned aspects and the description of the Figure 2 in the Discussion section:

“Under normal circumstances as has been portrayed in Figure 2, aproximatively 90% of the total oxygen is taken up by the mitochondria in order to produce ATP, ROS resulting through the partial four-electron reduction of O2 to H2O [11].

When the metabolic rate is increased, this is defined by a high production of ROS and is reflected by a depletion of the main antioxidant enzymes, in the present case, SOD and GPx [12]. However, the only reliable possibility to decrease ROS generation is to uncouple the metabolic rate [13].

Regardless the status of the organism, ROS gradually promotes a dysbiosis [14], apart from the imparative mitochondrial dysfunction which exacerbate ROS production [15]. Once the eubiosis is lost, a hazardous environment within the gut microflora emerges and disturb its entire metabolism [16,17].

Subsequently are activated a series of pro-inflammatory cytokines which further alter the protein metabolism [18]. Accordingly, the levels of MDA are increased, NH3 is released and through the portal vein circulate to the liver [2].

NH3 enters the systemic circulation and when the levels are higher than 500 µM, it occurs a systemic inflammation of the glial cells, the blood-brain barrier being unable to act efficiently which ultimately leads to hepatic encephalopathy [19].

It must be taken into considerations that this mechanism applies by excluding all harmful exogenous factors.”

Thank you!

Figure 2. Schematic representation on how oxidative stress gradually disrupts the homeostasis and promote hepatic encephalopathy. Given the fact that humans are strictly aerobic organisms, ROS results through the partial four-electron reduction of O2 to H2O. When the metabolic rate is increased, this is defined by a high production of ROS and vice versa and is reflected by a decreased of SOD and GPx. ROS promotes a dysbiosis, creating a hazardous environment in the gut microflora which disturb its entire metabolism. Subsequently are activated a series of pro-inflammatory cytokines which further alter the protein metabolism. Accordingly, it occurs the accumulation of MDA, NH3 is released and through the portal vein circulate to the liver. NH3 enters the systemic circulation and when the levels are higher than 500 µM, a system inflammation of the glial cells is installed. The blood-brain barrier no longer is able to act efficiently and ultimately leads to hepatic encephalopathy. It must be taken into considerations that this mechanism applies by excluding all harmful exogenous factors.

Additional comments

- there is no threshold for "high ammonemia" (this can vary according to the lab). Authors should precise this parameter that define their groups

Response: The threshold of high ammonemia was any value above the normal threshold of the hospital laboratory 30-120 mcg/dl . In this regard, the normal ammonemia was considered any value below 120mcg/dl and high ammonemia all values above 120mcg/dl

- a description of spontaneous portosystemic shunts may help to explain non hyperammonemic encephalopathy

Response: This is an interesting and pertinent remark, but, unfortunately, this was beyond the purpose and capabilities of our study which could not include the evaluation of patients in this regard. Still, we did add it to the limitations of our present study and it will be part of our future plans. Thank you!

- furthermore, the authors should discuss more thoroughly this group of non-hyperammonemic encephalopathy in the light of their findings- in this view, the discussion is too long and repeats some of the data detailed in the introduction and concentrate on the previous comments.

Response: Some of the paragraphs that included repetitive statements were eliminated From the Discussion section. Also, as you kindly suggested, we did add the additional Figure 2 and the relevant explanations attached to this figure. Thank you!

Round 2

Reviewer 1 Report

In general, the previous concerns and comments have been addressed appropriately.